# Mechanism and Influence Factors of Abrasion Resistance of High-Flow Grade SEBS/PP Blended Thermoplastic Elastomer

**DOI:** 10.3390/polym14091795

**Published:** 2022-04-28

**Authors:** Shuwen Liu, Jun Qiu, Lili Han, Xueyan Ma, Wenquan Chen

**Affiliations:** 1College of Material Science and Engineering, Tongji University, Shanghai 200082, China; shuwen_liu@patac.com.cn; 2Pan Asia Technical Automotive Center Co., Ltd., Shanghai 201201, China; 3Shandong Dawn Polymer Material Co., Ltd., Yantai 265700, China; hanlili022225@foxmail.com (L.H.); mxy@chinadawn.cn (X.M.)

**Keywords:** automotive interior, injection molding soft skin, abrasion resistance, high-flow thermoplastic elastomer, hydrogenated styrene-butadiene-styrene block copolymer

## Abstract

Hydrogenated styrene-butadiene-styrene block copolymer (SEBS)/polypropylene (PP) blended thermoplastic elastomer (TPE) is suitable for preparing the automotive interiors because of its excellent elasticity, softness, weather resistance, low odor, low VOC and other environmental-friendly properties. The skin of the automobile instrument panel is an appearance part, which requires excellent friction loss resistance of surface. In this paper, the high-flow SEBS/PP blended thermoplastic elastomer (TPE) suitable for the preparation of injection molding skins for automobile instrument panel was studied. By comparing the Taber abrasion and cross-scratch properties, the effects of SEBS’s molecular weight, styrene content in the molecule, molecular structure and types of lubricating agents on the friction loss properties of the material were investigated. The results show that under the same SEBS molecular structure, the higher the molecular weight within a certain range, the better the wear resistance of high-flow SEBS/PP type TPE, but the ultra-high molecular weight exhibits lower wear resistance than high molecular weight; The high-flow SEBS/PP blended TPE prepared by medium styrene content SEBS has better abrasion resistance; TPE prepared by star SEBS is better than linear SEBS; Adding silane-based lubricating agents is beneficial to improve the friction loss resistance of the material, especially combined use of high and low molecular weight silicone.

## 1. Introduction

As a component used in a relatively closed interior space, the soft skin of the automobile instrument panel requires not only excellent appearance quality and comfortable touch, but also low emission such as low odor, low VOC, and low fogging [1,2,3]. Additionally, as an automotive material for outdoor application, it is required to exhibit excellent aging resistance and weather resistance. Because of covering the blasting part of the airbag, the skin should have excellent high and low temperature resistance, and excellent toughness in the using environment to ensure personal safety at the moment of airbag explosion. The traditional automobile interior skin of automobile instrument panel mainly is divided into two categories: polyvinyl chloride (PVC) slush skin and thermoplastic polyolefin (TPO) vacuum-molded skin [4]. PVC cannot meet the requirements of people’s increasing demands on life quality, health and low emission because of its poor aging resistance, poor high and low temperature resistance, high level emission such as high odor, high VOC and high fogging level, unfriendly driving health, and difficulties to be recycled [5,6,7,8]. Other disadvantages of PVC slush skin also include the long processing period, high energy consumption and high mold cost for regular replacement of sub-mold [9]. The vacuum molded TPO skins exhibit the advantages of low odor and low VOC, but it requires multiple processes such as extrusion calender, vacuum molding and spray coating, which means complex processing technology and a long processing period.

With excellent elasticity, aging resistance, weather resistance, high and low temperature resistance and recyclability, hydrogenated styrene-butadiene-styrene block copolymer (SEBS)/polypropylene (PP)-blended thermoplastic elastomer (TPE) can meet the extremely high requirements of automotive interiors for excellent touch and low emission of automotive products [10,11,12,13,14,15]. What is more, the hardness and fluidity of the product can be designed in a wide range, and the touch feeling is excellent, so it is gradually favored in the application of automotive interiors [16,17]. Nonpolar characteristics of SEBS and PP provide 10^15^~10^16^ Volume resistivity, meaning excellent insulation resistivity, which protects the driver and passenger from leakage of electricity at extreme conditions such as serious traffic accidents.

When injection molding process was used to form the soft skin of the automobile instrument panel, low wall thickness (0.9~1.2 mm) and large area of interior skin require extremely high melt flow rate. At the same time, the interior skin inevitably comes into contact with sharp objects such as nails and keys, so the product is required to perform good wear resistance and scratch resistance. Non-polar characteristics of SEBS and PP lead to low cohesion force between molecules. For ultra-high flow rate required for thin-wall injection molding, traditional methods are realized by using low-molecular-weight SEBS or filling a large amount of low-viscosity paraffin oil into high-molecular-weight SEBS [18]. However low molecular weight or a large amount of paraffin oil is filled between molecular chains, causes lower cohesion force between SEBS molecules, and leads to poor wear resistance, and failure is prone to occur during friction and wear. The elastomer material is soft, and the contact area is prone to large deformation under compression force, so it is easy to form a “wrapping” effect on the object applying friction. That is, it is in closer contact between TPE skin and object applying friction, which cause the friction coefficient to increase and the significant frictional heat to generate and Accelerate wear failure. At present, when traditional TPE interior skin prepared by injection molding was applied in some Japanese cars, paint-spray was required on the surface of the skin to meet the requirements of wear resistance. Studies have also shown that the addition of TPEE or TPU can improve the wear resistance of the material [19], but meanwhile the emission of odor and VOC will also increase significantly. As mentioned above, low odor and VOC emission are required for automotive interior materials, so this method is not suitable.

Previous studies have shown that SEBS with higher molecular weight means stronger molecular chain entanglement especially for SEBS with star molecular structure, which results in higher surface wear resistance of TPEs prepared with them. SEBS with higher styrene content in the molecular chains has stronger conjugation force between chains, and TPEs prepared by them also exhibit better wear resistance [20,21,22]. However, all of those studies were only for TPEs with common fluidity. No studies about the influence factors on abrasion resistance of TPEs with ultra-high flow rate exceeding 50 g/10 min (tested at 190 °C × 2.16 kg) can be found. In this paper, PP with higher flow rate was prepared and applied to TPE formula to ensure the high flow rate, and the effects of the molecular weight of SEBS, content of styrene in the molecular chain, molecular structure and the lubricating agents and masterbatches on the friction loss resistance of the material were mainly studied. For composite materials, the wear mechanisms are mainly abrasive wear, adhesive wear, and fatigue wear. In practical applications, the wear of materials is usually in a composite form of two or even three kinds of abrasion at the same time, aiming at the practical application of automobile interior skin. In this paper, Taber abrasion test and cross scratch test was used to study abrasion resistance of injection molded skins.

## 2. Experimental Materials and Methods

### 2.1. Main Raw Materials and Equipment

SEBS G1650, G1654, G1651, G1633: American Kraton, commercially available; SEBS YH-511: China Petroleum & Chemical Corporation (Yueyang, China), commercially available; PP 225 (powder): Zhejiang Hongji Petrochemical Co., Ltd. (Jiaxing, China), commercially available; Di-tert-butyl peroxide DTBP: AkzoNobel Chemicals (Tianjin, China) Co., Ltd., commercially available; PP CB5290: Korea Petrochemical IND, Co., Ltd. (Ulsan Metropolitan City, Korea), commercially available; Paraffin oil KP6030: PetroChina Karamay Petrochemical Co., Ltd. (Karamay, China), commercially available; Lithium dodecyl stearate 18K: Katsuta, Japan, commercially available; Silicone L5-4 (Extra high molecular weight, Mw is about 700,000, Hereinafter refer to as HWSi): Wacker Chemical Group Co., Ltd. (Berghausen, Germany), commercially available; Silicone oil PMX-200 (Low Molecular weight, Mw is about 10,000, Hereinafter refer to as LWSi): Dow Corning (Zhangjiagang, China) Co., Ltd., commercially available; POE 8180: DOW Chemical, commercially available; Silicone masterbatch HG-650: Zhejiang Java Speciality Chemicals Co., Ltd. (Shaoxing, China), commercially available; Polytetrafluoroethylene JTC-308: Zhejiang Juhua Co., Ltd. (Quzhou, China), commercially available.

High-speed mixer: SHR-10A, Guangdong Xieda Machinery Co., Ltd. (Dongguan, China); Twin-screw extruder: SHJ-35, Nanjing Juli Chemical Machinery Co., Ltd. (Nanning, China); Injection molding machine: UN120SK, Yizhimi Precision Machinery Co., Ltd. (Foshan, China); Shore A hardness tester: ZWICK ARMATUREN GMBH (Ulm, Germany); Melt index meter: MF30, CEAST, Italy; Universal tensile testing machine: 3366, Instron Engineering Corporation (Boston, MA, USA); Taber Abrasion Tester: 1755, Taber Industrie (New York, NY, USA); Cross scraping instrument: 430P-1, ERICHSEN INC. (Ann Arbor, Germany); Two-position imager: SV4030, Dongguan Tianqin Instrument Co., Ltd. (Dongguan, China).

### 2.2. Preparation of Samples

#### 2.2.1. Preparation of Ultra-High Flow Rate PP (Hereinafter Refer to as PP-UHMFR)

PP 225 power and DTBP were mixed in a high-speed mixer at middle speed for 4 min according to 1000:3 weight ratio, and then blended and extruded by a twin-screw extruder at a temperature 230~270 °C, followed by being granulated into pellets with a plastic granulator. Additionally, an Ultra high melt flow rate up to about 1500 g/10 min (230 °C × 2.16 kg) PP was prepared.

#### 2.2.2. Preparation of Masterbatch Contains HWSi and LWSi

POE 8180, Silicone L5-4 and Silicone oil PMX-200 were mixed in a high-speed mixer at a high speed for about 8 min according to 50:45:5 weight ratio, and then blended and extruded by a twin-screw extruder at a temperature 150~200 °C, followed by being granulated into pellets with a plastic granulator. Additionally, a silicone masterbatch containing HWSi and LWSi was prepared (Hereinafter refered to as M-H/LSi).

#### 2.2.3. Preparation of SEBS/PP TPEs and Samples

The paraffin oil KP6030 and SEBS were mixed in a high-speed mixer at a high speed for 5 min according to a certain weight ratio, and then left to stand for 30 min to ensure the full absorption of the KP6030 oil to obtain oil-extended SEBS (O-SEBS); the O-SEBS, PP and additives were weighed in proportion and mixed thoroughly; the mixture were granulated by a twin-screw extruder, injected injection machine and mold into specific samples.

### 2.3. Testing and Characteristics

The hardness was tested according to the ISO 868:2003 standard, and the recording time was 15 s; The melt flow rate (MFR) which was used to characterize the fluidity of materials was tested according to the ISO 1133-1:2011 standard at 190 °C × 2.16 kg; Higher MFR means better fluidity; The tensile performances were tested according to the ISO 37:2017 standard, and the tensile speed was 500 mm/min, the test results in this paper are the average values after 5 tests, and standard deviation σ was used to characterized variances of tensile strength and elongation at break of those 5 tests; The tear strength was tested according to ISO 34-1:2015 standard; The Taber abrasion was tested according to ISO 5470-1-2016 standard, the load was 0.5 kg, 1000 r. Appearance Rating was evaluated according to GMW 3208-2017, Rating 10 is the best and Rating 2 is the worst; The cross scratch was tested according to the GB/T 9286-2015 standard, the load was 10N, and the speed was 1000 mm/min.

## 3. Result and Discussion

### 3.1. Influence of SEBS Molecular Weight on Mechanical Properties and Wear Resistance of High-Flow SEBS/PP TPE

The formulas of linear SEBS with similar styrene content and molecular chain structure, and various molecular weights were applied to prepare high-flow SEBS/PP TPE suitable for injection molding skin are listed on Table 1.

Table 2 shows the mechanical properties of high-flow TPEs prepared by SEBS with similar styrene content and molecular chain structure but various molecular weights. As shown in Table 2, the hardness of high-flow TPEs prepared by SEBS with similar styrene content and molecular chain structure but various molecular weights are equivalent. When the molecular weight of SEBS is low, the mechanical properties of the obtained high-flow TPE are low because the intermolecular cohesive force of SEBS is so weak that the molecular chain slippage occurs easily. Higher molecular weight of SEBS lead to higher mechanical properties of high-flow TPE prepared. However, the increase in molecular weight followed by the increase in intermolecular cohesion and the increase of the entanglement point between molecular chains, therefore the fluidity of the prepared high-flow TPE decreases.

Table 3 is the cross-scratch value corresponding to Table 1. Figure 1 and Figure 2 are respectively the Taber abrasion test photos before and after the test corresponding to Table 1. Although there is a slight difference in ΔE after cross-scratching, as shown in Figure 1, it can be seen from the test-piece that the high-flow TPEs made of SEBS with various molecular weights have similar scratch resistance. There is only slight scratch without whitening phenomenon and visible differences between various samples.

Table 3 shows the abrasion resistance of TPE prepared by SEBS with similar structure and various molecular weights. It shows that TPE-HW exhibits the least mass loss of materials after Taber-abrasion, with Rating 4.5. However, the difference of ΔE value after cross-scratch was not obvious.

Figure 2 shows photos of the test samples after Taber abrasion of TPE prepared by SEBS with various molecular weights. It can be seen from Figure 2 that the Taber abrasion resistance is as follows: TPE-LW < TPE-MW < TPE-UHW < TPE-HW, namely SEBS G1654 < G1650 < G1651 < G1633. Using a two-position imager to magnify the wear scratch by 40 times (as shown in Figure 3):

Figure 3 is a magnified view of the abrasion scratch of the sample in Figure 2 after the Taber abrasion test, and the bottom left one (e) is the magnified view before Taber abrasion test. As shown in Figure 3, compared with the samples before Taber abrasion test, the convex part of the grain of the TPE-MW wear test sample has been completely damaged, and stuck to the next convex part of the grain, and the interface of the convex and concave part has been unclear; Although the convex part of the grain of TPE-HW is slightly damaged, the interface between the convex and concave part of grain is clear; Slight damage in the convex of grain can be observed and the interface between convex and concave part is slightly blurred in TPE-UHW sample after Taber abrasion test. It can be inferred that TPE-HW exhibits the best abrasion resistance, that is, the high flow TPE made by G1651 has the best abrasion resistance.

According to Figure 3, it can be inferred that SEBS has similar molecular chain structure and styrene content, because of more entanglement points between molecular chains in SEBS with high molecular weight than SEBS with low molecular weight, and cohesion force between molecular chain is stronger than SEBS with low molecular weight, which causes higher abrasion resistance of surface. In addition, for SEBS with various molecular weight, mechanical performances reduce faster while temperature increases for SEBS with low molecular weight than with high molecular weight, therefore, the surface of the sample is more easily damaged due to temperature increase by heat generation during Taber abrasion test. The abrasion resistance is G1654 < G1650 < G1651 and increases sequentially with the increase in the molecular weight. When the molecular weight is low, the thermal resistance of the material is poor. During the abrasion test, as the temperature of the sample increases because of abrasion heat generation, the material softens and becomes sticky, and fragments come from the sample during abrasion test is therefore easy to adhere on the surface of the grinding wheel of Taber abrasion tester to form a transfer film on the surface of the wheel, which reduces the roughness and friction coefficient of the grinding wheel surface and reduces wear. The ultra-high molecular weight SEBS (TPE-G1633 in Figure 3) exhibits high thermal stability, and the temperature increase from abrasion heat generation is not enough to make the fragment from abrasion fully adhere to the surface of the grinding wheel to form a relatively dense “transfer film”. The surface of the test sample is always rubbed directly by the rough grinding wheel surface, so exhibits weaker abrasion resistance than TPE with high molecular weight SEBS. Therefore, G1651 showed the best abrasion resistance in the above test, better than G1633 with the highest molecular weight.

As discussed above, cross scratch is a test method of one-time scratch test which is completed instantly, and abrasion heat generation is not enough to obviously influence on wear level, so TPEs with similar hardness exhibit wear level in cross scratching test. However, Taber abrasion test is a relatively long-lasting continuous abrasion. The abrasion heat generation is an obvious influence on the wear level of the material during long-time continuous abrasion test. Abrasion heat generation accelerates wear damage, especially during long-time continuous abrasion.

### 3.2. Effect of Styrene Content on Mechanical Properties and Abrasion Resistance of High-Flow TPE

Table 4 shows the formulas for preparing high-flow TPE from SEBS with various styrene contents. In the high-flow TPE made of SEBS with similar molecular weight, similar molecular chain structure and various styrene content, SEBS with high styrene content has a higher content of hard segments in the molecular chain. Therefore, under the same proportion in the formula, the high-flow TPE prepared by SEBS with high styrene content exhibits higher hardness than with low styrene content. To obtain high flow TPE prepared by high styrene SEBS with the same hardness, it is necessary to reduce the proportion of PP in the formula. For example, when G1535 is used in Table 4, the addition amount of CB5290 is reduced from 10% to 8%. For high styrene content SEBS with low molecular weight such as YH-511, the hardness of high flow TPE made by them does not increase much. Therefore, to obtain high-flow TPE with similar hardness, the ratios in the formulas can be kept unchanged.

Table 5 shows the mechanical properties of high-flow TPE prepared by SEBS with various styrene content. In theory, SEBS with higher styrene content can form a stronger conjugated structure because it contains more benzene rings, which enhance cohesion between molecular chains, and exhibits higher mechanical properties and wear resistance. However, as shown in Table 5, in fact, the mechanical properties of high-flow TPE prepared by high styrene content SEBS are lower and MFR is higher than those by medium styrene SEBS. Although the strength of high-flow TPE prepared by SEBS with medium and high molecular weight with high styrene content is low, the fluidity is higher. When the molecular weight is similar, the content of ethylene-butene structural units that can form entanglements between molecular chains in SEBS with high styrene content is low, and the degree of molecular entanglement decreases, resulting in a decrease in the melt viscosity of the material and an increase in fluidity. The molecular weight decreases followed by and increase in fluidity. As the content of styrene increases, the content of ethylene-butene in SEBS that has good compatibility with polypropylene decreases, and the compatibility of SEBS with polypropylene becomes poor, and the interface bonding force between SEBS and PP decreases. The interface between SEBS and PP is more easily damaged under external force, which results in poor mechanical properties of the obtained high-flow TPE.

Table 6 shows the abrasion resistance of TPE prepared by SEBS with various styrene content. It shows that TPE-Ms/Hw prepared by SEBS with middle Styrene content and high molecular weights exhibits the least mass loss of materials after Taber-abrasion, with Appearance rating 4.5, while that of the other two are both 3.5. However, the ΔE value after cross-scratch did not show obvious difference.

Figure 4 and Figure 5 show the high-flow TPE samples prepared by SEBS with various styrene content after the cross-scratch and Taber test, respectively. As shown in Figure 4 and Figure 5, the high-flow TPEs prepared with high styrene content SEBS exhibit poor abrasion resistance in Taber and cross-scratch test. The abrasion resistance of TPE made by high styrene content such as G1535 with high molecular weight or YH-511 with medium molecular weight is not as good as that of G1651 with medium styrene content and high molecular weight. High styrene content SEBS and PP are poor in compatibility, and the interfacial bonding force between SEBS and PP is low. Interface of PP and SEBS are prone to delamination during abrasion test.

The samples of high-flow TPE prepared by SEBS 1535 and SEBS YH-511 after Taber abrasion were observed by magnifying 40 times, as shown in Figure 6a,b, the dermatoglyphic epidermis of both are partially displaced. From the cross-sectional inspection of the displacement position in Figure 6, as shown in Figure 6c,d, some floating skins can be observed on the left side of the two figures, and the material has been delaminated. It can be speculated that the compatibility of the material is poor, and there are large-sized dispersed phases in the TPE, which result in the strength reduction in the material surface and the deterioration of the abrasion resistance (consistent with Figure 5, that is, the wear resistance of G1535 and YH-511 are not good).

### 3.3. Influence of SEBS Molecular Chain Structure on the Properties of High-Flow TPE

There are two types of morphological structures of SEBS molecule: linear and star. In the case of a certain molecular weight, Star-SEBS exhibits better abrasion resistance and elasticity than linear SEBS. Because types of star-SEBS which can be obtained in the market is limited, only two types of star-SEBS with different molecular weight for research are used in this paper.

Larger mean square end-to-end distance between molecular chains exists in linear SEBS with the similar molecular weight, which easily forms more physical entanglement points. The physical cross-linking network is better than the star-SEBS, and mechanical properties are higher. Therefore, the mechanical properties of the high-flow TPE prepared by the linear G1651 are the best overall. The star-shaped SEBS contains more diblock SEBs, which can be regarded as plasticizers. Therefore, the fluidity of high-flow TPE prepared by star-SEBS of the same molecular weight grade is slightly improved, so the melt flow rate of TPE prepared by YH-602T in Table 7 is slightly higher than that of G1651. However, for ultra-high molecular weight star-SEBS, such as YH-604T, it shows a much higher physical entanglement network than that of ordinary linear SEBS, poor fluidity and poor processing performance because of its high molecular weight. When melt blends with high fluidity PP, the compatibility is poor, resulting in poor fluidity and mechanical properties of product. In this paper, two types of star-SEBS with different molecular weights specifications were used for research. The formulas of TPE are shown in Table 8.

Table 9 shows the comparison of the abrasion resistance of TPE prepared by SEBS with various molecular structures. Figure 7 and Figure 8 are the corresponding test pieces after cross scratch and Taber abrasion.

Figure 9 is a photo magnified 40 times of the samples after Taber-abrasion test in Figure 8 magnified 40 times. Although the ΔE of the cross-scratch is low (as showed in Table 9), it can be seen from the samples after cross-scratch test that a certain degree of burr can be observed on TPE-HWS surface, which does not occur in the TPE-LWS sample (as shown in Figure 7). As shown in Figure 8 and Figure 9, TPE-LWS has the best wear resistance and scratch resistance, followed by TPE-HWL, and TPE-HWS is the worst. The speculated reason for the unclear grain interface of TPE-HWS is that the molecular weight of YH-604T is so high that it is difficult to be plasticized, which results in poor compatibility of PP and SEBS during shot time processing by twin-screw extruder, therefore the interface force between PP and high molecular weight star-SEBS is weak, and the surface strength of the material is poor. At the same time, the molecular weight is too high, and the fragments from abrasion are not enough sticky enough to form a dense transfer film on the surface of the grinding wheel, resulting in poor Taber abrasion performance and high wear degree. Compared with SEBS YH-604T, the molecular weight of YH-602T is slightly lower, and the compatibility is better, the physical cross-linking network of star structure is stronger than linear, therefore high flow TPE prepared by YH-602T exhibits the best abrasion resistance.

### 3.4. Influence of Lubricating Agents and Masterbatches on the Properties of High-Flow TPE

By adding different lubricating agents in the formula, the friction coefficient of the material can be effectively reduced, the frictional force and the abrasion heat generation can be reduced, followed by abrasion resistance of the material which can be improved significantly [23]. Common lubricants include siloxanes, polytetrafluoroethylene, polyethylene wax, stearate, amides, etc. [24]. Among them, amide lubricants such as erucic acid amide and oleic acid amide can quickly migrate to the surface and form a smooth layer to reduce the friction coefficient and abrasion heat generation, which help to reduce wear damage of material. However excessive addition would cause poor appearance such as whitening and poor adhesive with PU sponge in the automobile instrument panel because of excessive migration. The addition of macromolecular organic additives containing fluorine or silicon can improve the abrasion resistance of the material and avoid surface sticky and whitening. Polytetrafluoroethylene (PTFE) can improve the abrasion resistance of materials to a certain extent due to its low friction coefficient [25]. PTFE is a kind of self-lubricating materials and exhibits low film coefficient. The addition of PTFE reduces friction of sample surface. so as to achieve the effect of increasing wear resistance. Liu Xiaoyan et al. [26] found that the friction coefficient of the material can be reduced by the addition of antistatic agents when they studied glass fiber. In this paper, the effects of silicone masterbatch, PTFE and lithium dodecyl stearate 18K on the wear resistance of high-flow TPE were studied.

For the study of the lubricating agents and masterbatches, only the lubricating agents and masterbatches is variable, and the formulas of other components remain unchanged, formula details can be seen in Table 10. The various lubricating agents or masterbatches are added.

Table 11 shows the formulas of TPE adding various lubricating agents or masterbatches. Which are based on the formulas in Table 10 with only differences in lubricating agents or masterbatches.

Figure 10 and Figure 11 respectively show the abrasion scratch of the samples with various lubricating agents and masterbatches after Taber abrasion and cross scratching. As shown in Figure 10 and Figure 11, the Taber wear can reach grade 8 by adding SR-100B, 18K and M-H/LSi. However, it can be seen from the pictures that obvious migration and whitening occurs on the surface of TPE-18K sample, and the surface is greasy, which cannot meet the requirements for practical interior applications. Antistatic agent has the effect of increasing wear resistance and self-lubricating, but low molecule and incompatibility with TPE of 18K cause easy migration to the surface of material easily. SR-100B and M-H/LSi are both silicone masterbatches which significantly exhibit the effect of improving friction loss of materials. By forming a lubricating layer on the surface of the TPE material, silicone can obviously reduce the friction coefficient and improve wear resistance of TPE. The M-H/LSi exhibits higher smoothness, more significant abrasion resistance improvement, and the actual feel is better than SR-100B, because M-H/LSi masterbatch contains both silicone with ultra-high molecular weight which exhibits excellent abrasion resistance and silicone oil with low molecular weight which exhibits low friction coefficient and heat generation. The research on the above lubricants shows that the addition of silicone masterbatch can significantly improve the anti-scratch wear performance of the material, especially the combined use of high and low molecular weight silicone, and the Taber abrasion can reach grade 8.

Table 12 shows the wear resistance of TPE prepared by various lubricating agents and masterbatches. As can be seen from Table 12, there is not obvious difference in the mass loss of materials after Taber Abrasion by adding various lubricating agents, but the appearance rating is mostly between level 6.0 and level 8.0, while only level 4.5 without lubricant. It also can be observed that Taber-abrasion can be significantly improved by adding lubricating agents and masterbatches. In addition, cross-scratch was significantly improved by adding lubricating agents or masterbatches, and the ΔE value could be reduced from 0.61 to 0.21. The TPE-SR100B and TPE-H/LSi perform better.

## 4. Conclusions

In this paper, when preparing high-flow SEBS/PP type TPE suitable for injection molding elastomer skin of automobile instrument panel, the characteristics of SEBS and the influence of lubricating agents and masterbatches on wear resistance were mainly studied. By comparing Taber abrasion and cross scratch properties, the following conclusions can be drawn:(1)In the case of similar molecular structure and styrene content of SEBS, high flow TPE prepared by higher molecular weight within a certain range can impove the abrasion resistance (such as G1651), but TPE with the ultra-high molecular weight SEBS (such as G1633) also exhibits worse wear resistance.(2)The wear resistance of high-flow TPE prepared by medium styrene content SEBS is better.(3)The high-flow TPE made of star-shaped SEBS has the most excellent wear resistance.(4)Adding siloxane-based lubricating agents and masterbatches is beneficial to improve the wear resistance of the material, especially with the combined addition of high and low molecular weight silicone, which was pre-dispersed into POE elastomer.(5)Mechanical performance and abrasion resistance is less affected by frictional heat generation for TPE prepared by SEBS with higher heat resistance. Large distance between the ends of molecules is benefit to abrasion resistance.(6)Good compatibility between SEBS and PP is benefit to coefficient reducing and abrasion resistance.

## Figures and Tables

**Figure 1 polymers-14-01795-f001:**
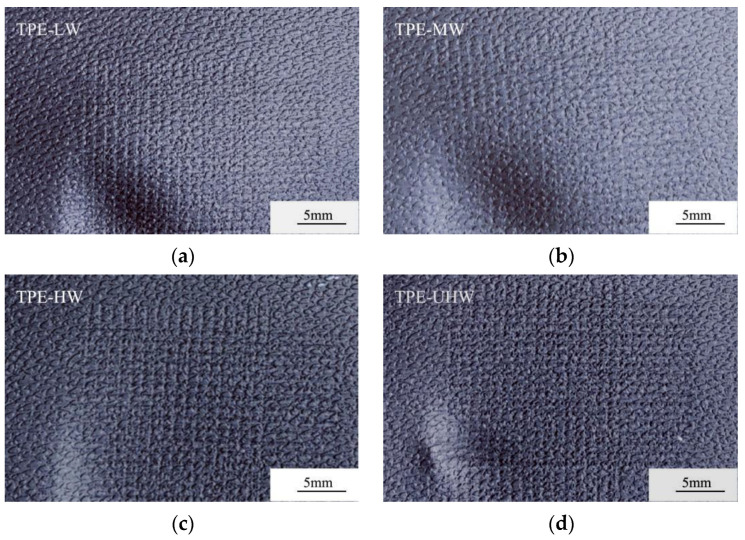
Samples after cross-scratch test of TPE prepared by SEBS with similar structure and various molecular weights. (**a**) TPE-LW; (**b**) TPE-MW; (**c**) TPE-HW; (**d**) TPE-UHW.

**Figure 2 polymers-14-01795-f002:**
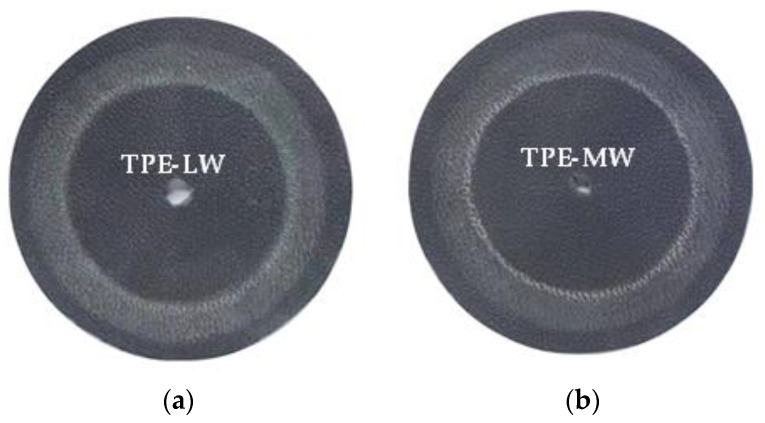
TPE samples after Taber abrasion test of TPE prepared by SEBS with similar structures and various molecular weight. (**a**) TPE-LW; (**b**) TPE-MW; (**c**) TPE-HW; (**d**) TPE-UHW.

**Figure 3 polymers-14-01795-f003:**
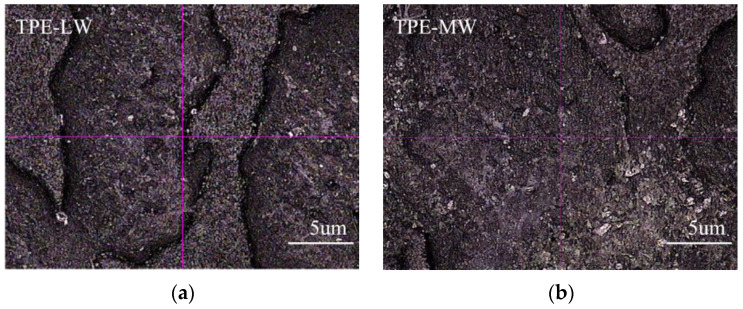
40 times magnification of the wear scratch of samples with similar structures and various molecular weight SEBS after Taber test. (**a**) TPE-LW; (**b**) TPE-MW; (**c**) TPE-HW; (**d**) TPE-UHW; (**e**) the magnified view before Taber abrasion test.

**Figure 4 polymers-14-01795-f004:**
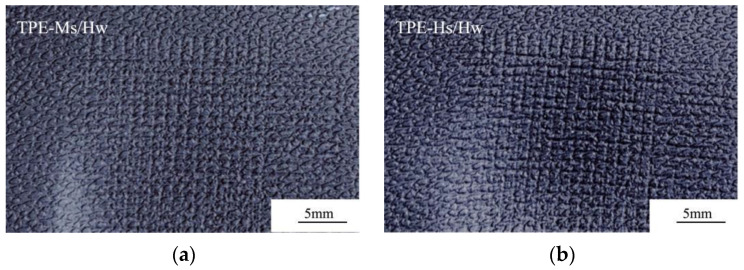
Comparison of TPE cross-scratched samples prepared from SEBS with various styrene contents. (**a**) TPE-Ms/Hw; (**b**) TPE-Hs/Hw; (**c**) TPE-Hs/Lw.

**Figure 5 polymers-14-01795-f005:**
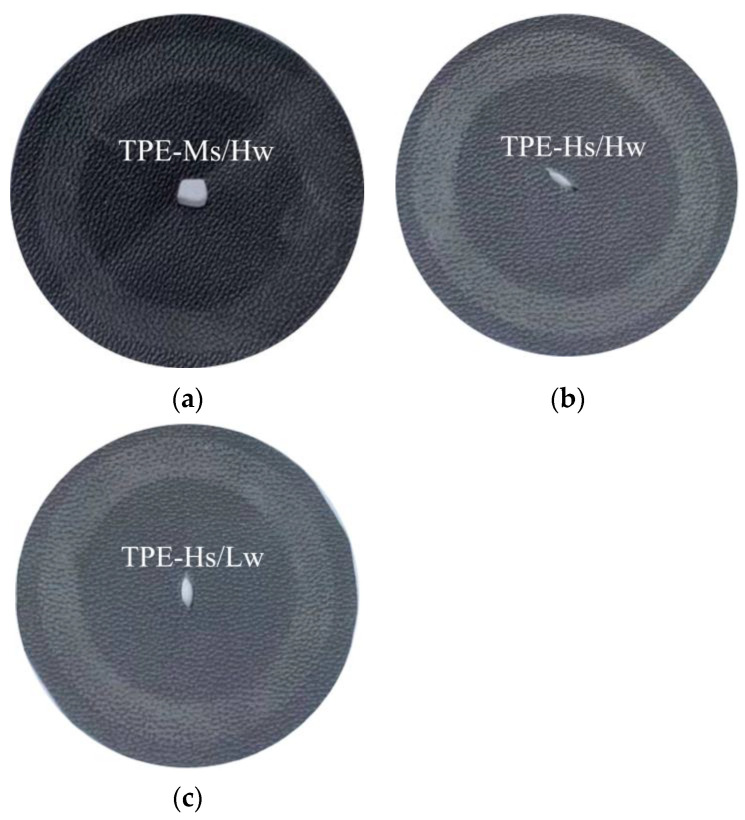
TPE samples prepared by SEBS with various styrene content after Taber abrasion. (**a**) TPE-Ms/Hw; (**b**) TPE-Hs/Hw; (**c**) TPE-Hs/Lw.

**Figure 6 polymers-14-01795-f006:**
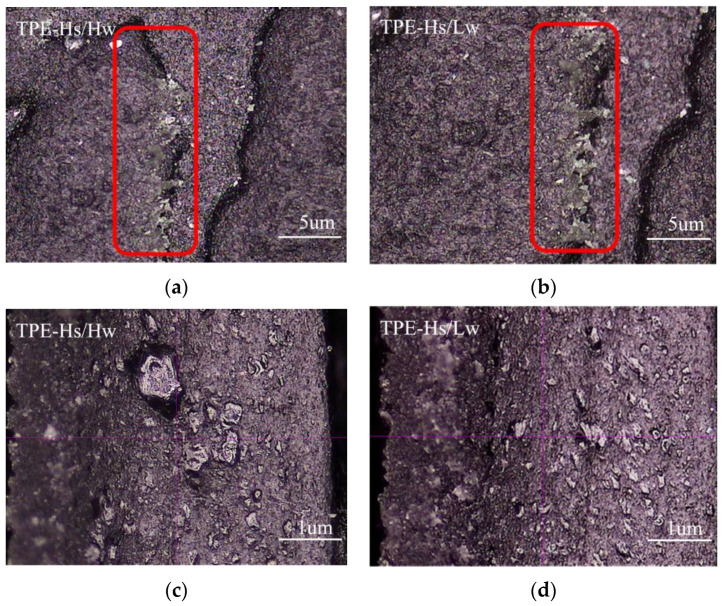
Comparison of abrasion scars of TPE with high styrene content. (**a**,**b**) are the pictures of TPE-Hs/Hw and TPE-Hs/Lw magnified 40 times respectively, (**c**,**d**) are partial magnifications of (**a**,**b**) with a magnification of 200 times).

**Figure 7 polymers-14-01795-f007:**
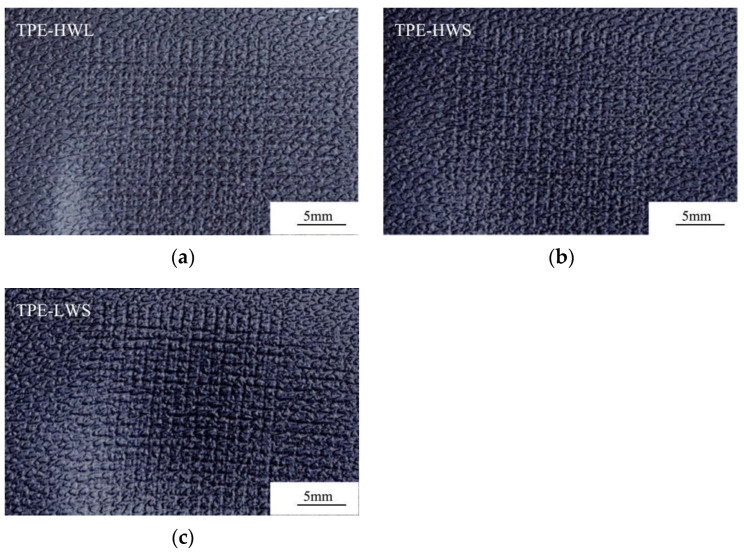
Cross scratching sample of TPE prepared by SEBS with various molecular structures. (**a**)TPE-HWL; (**b**) TPE-HWS; (**c**) TPE-LWS.

**Figure 8 polymers-14-01795-f008:**
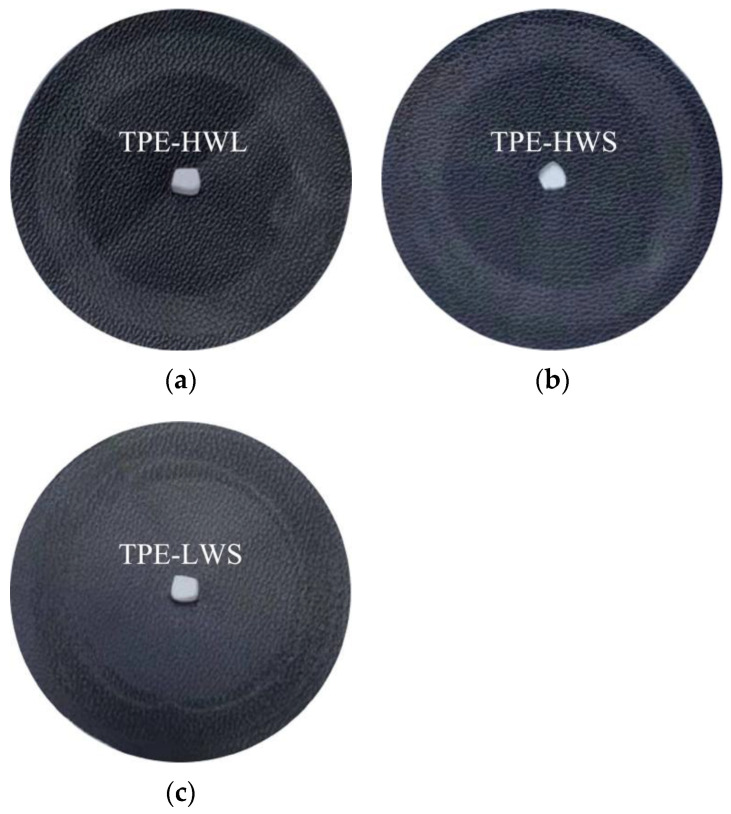
TPE samples prepared by SEBS with various molecular structures after Taber abrasion. (**a**) TPE-HWL; (**b**) TPE-HWS; (**c**) TPE-LWS.

**Figure 9 polymers-14-01795-f009:**
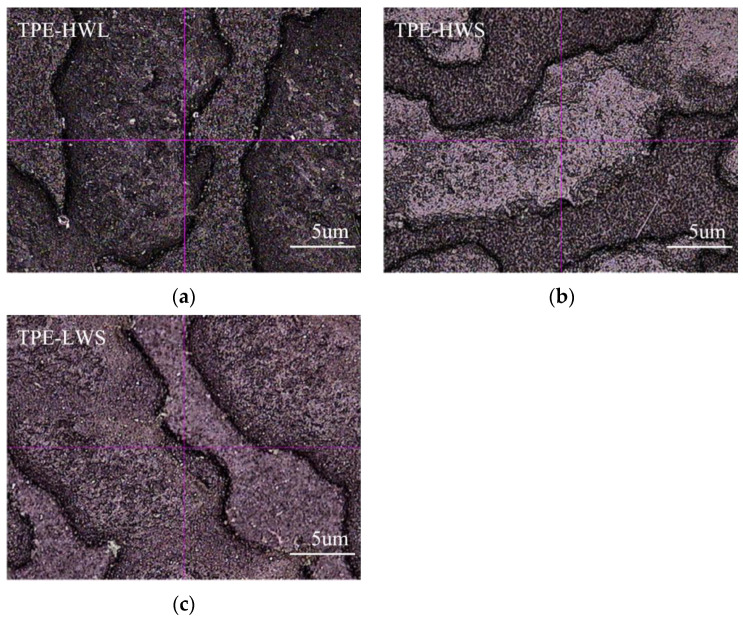
40 times magnified photos of samples prepared by SEBS with various molecular structures after Taber-abrasion test. (**a**) TPE-HWL; (**b**) TPE-HWS; (**c**) TPE-LWS.

**Figure 10 polymers-14-01795-f010:**
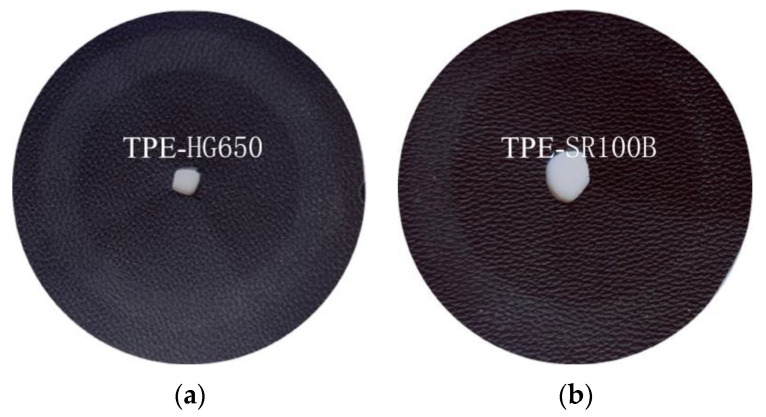
TPE samples prepared by various lubrication agents and masterbatches after Taber Abrasion. (**a**) TPE-HG650; (**b**) TPE-SR100B; (**c**) TPE-18K; (**d**) TPE-H/LSi; (**e**) TPE-PTFE.

**Figure 11 polymers-14-01795-f011:**
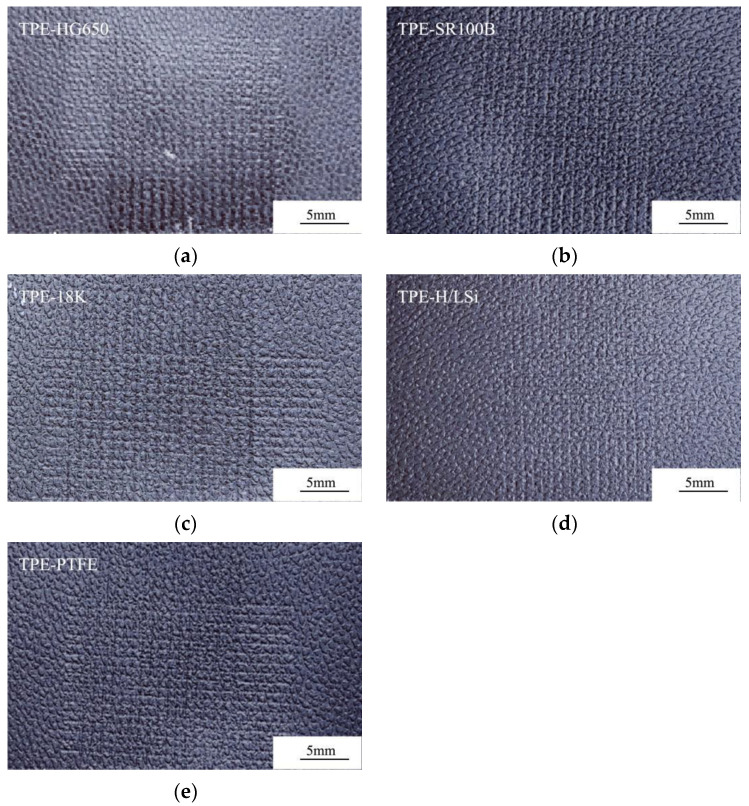
Cross scratch samples of TPE prepared by various lubricating agents and masterbatches. (**a**) TPE-HG650; (**b**) TPE-SR100B; (**c**) TPE-18K; (**d**) TPE-H/LSi; (**e**) TPE-PTFE.

**Table 1 polymers-14-01795-t001:** Formulas of TPE prepared by SEBS with similar structure and various molecular weights.

Sample	Content of Styrene (%)	Structural Type of the Molecule	SEBS MW Description	TPE-LW	TPE-MW	TPE-HW	TPE-UHW
G1654	30	Line	Low	40			
G1650	31	Line	medium		40		
G1651	31	Line	High			40	
G1633	30	Line	Ultra-high				40
PP-UHMFR				20	20	20	20
PP CB5290				10	10	10	10
KP6030				30	30	30	30
Black masterbatch				2	2	2	2
Other additives				0.3	0.3	0.3	0.3

**Table 2 polymers-14-01795-t002:** Mechanical properties of TPE prepared by SEBS with similar structure and various molecular weights.

Sample	Shore Hardness (A)	MFR (g/10 min)	Tensile Strength (MPa)/σ	Stress at 100% Strain (MPa)	Elongation at Break (%)/σ	Tear Strength (KN/m)
TPE-LW	75	140	6.7/0.37	2.4	530/11.9	25.2
TPE-MW	75	123	7.9/0.27	2.5	560/12.7	27.4
TPE-HW	76	91	8.2/0.61	2.7	751/17.04	34.2
TPE-UHW	77	59	10.2/0.63	2.7	732/10.28	37.5

**Table 3 polymers-14-01795-t003:** Abrasion resistance of TPE prepared by SEBS with similar structure and different molecular weights.

Sample	Taber Abrasion		Cross Scratch
	Mass Loss (mg)	Appearance Rating	ΔE
TPE-LW	0.35	3.0	0.75
TPE-MW	0.23	3.5	0.71
TPE-HW	0.08	4.5	0.61
TPE-UHW	0.10	4.0	0.60

**Table 4 polymers-14-01795-t004:** Formulas of TPE prepared by SEBS with various styrene contents.

Sample	Content of Styrene (%)	Structural Type of the Molecule	SEBS MW Description	TPE-Ms/Hw	TPE-Hs/Hw	TPE-Hs/Lw
G1651	31	Line	High	40		
G1535	51	Line	High		42	
YH-511	51	Line	Low			40
PP-UHMFR				20	20	20
PP CB5290				10	8	10
KP6030				30	30	30
Black masterbatch				2	2	2
Other additives				0.3	0.3	0.3

**Table 5 polymers-14-01795-t005:** Mechanical properties of TPE prepared by SEBS with various styrene content.

Sample	Shore Hardness (A)	MFR (g/10 min)	Tensile Strength (MPa)/σ	Stress at 100% Strain (MPa)	Elongation at Break (%)/σ	Tear Strength (KN/m)
TPE-Ms/Hw	76	91	8.2/0.61	2.7	751/17.04	34.2
TPE-Hs/Hw	76	125	5.6/0.22	2.8	504/13.08	27.4
TPE-Hs/Lw	76	125	5.7/0.28	2.7	489/11.9	27.2

**Table 6 polymers-14-01795-t006:** Abrasion resistance of TPE prepared by SEBS with various styrene content.

Sample	Taber Abrasion		Cross Scratch
	Mass Loss (mg)	Appearance Rating	ΔE
TPE-Ms/Hw	0.04	4.5	0.61
TPE-Hs/Hw	0.32	3.5	0.58
TPE-Hs/Lw	0.26	3.5	0.57

**Table 7 polymers-14-01795-t007:** Mechanical properties of TPE prepared by SEBS with various molecular chain structures.

Sample	Shore Hardness (A)	MFR (g/10 min)	Tensile Strength (MPa)/σ	Stress at 100% Strain (MPa)	Elongation at Break (%)/σ	Tear Strength (KN/m)
TPE-HWL	76	91	8.2/0.61	2.7	751/17.04	34.2
TPE-HWS	78	56	4.6/0.20	2.8	650/17.2	30.2
TPE-LWS	76	106	5.6/0.27	2.7	512/14.64	28.4

**Table 8 polymers-14-01795-t008:** Formula design of TPE prepared by SEBS with various molecular chain structures.

Sample	Content of Styrene (%)	Structural Type of the Molecule	SEBS MW Description	TPE-HWL	TPE-HWS	TPE-LWS
G1651	31	Line	High	40		
YH-604T	33	Star	Ultra-high		40	
YH-602T	35	Star	High			40
PP-UHMFR				20	20	20
PP CB5290				10	10	10
KP6030				30	30	30
Black masterbatch				2	2	2
Other additives				0.3	0.3	0.3

**Table 9 polymers-14-01795-t009:** Abrasion resistance of TPE prepared by SEBS with various molecular structures.

Sample	Taber Abrasion		Cross Scratch
	Mass Loss (mg)	Appearance Rating	ΔE
TPE-HWL	0.04	4.5	0.61
TPE-HWS	0.06	4.0	0.55
TPE-LWS	0.31	5.0	0.56

**Table 10 polymers-14-01795-t010:** Basic formulas of different lubricating agents and masterbatches.

SEBS G1651	PP-UHMFR	PP CB5290	KP6030	Black Masterbatch	Lubricating Agents and Masterbatches	Other Additives
40	20	10	30	2	2	0.3

**Table 11 polymers-14-01795-t011:** Formulas of various lubricating agents and masterbatches.

Sample	TPE-HG650	TPE-SR100B	TPE-18K	TPE-H/LSi	TPE-PTFE
Silicone masterbatch HG-650	2	\	\	\	\
Silicone masterbatch SR-100B	\	2	\	\	\
Lithium dodecyl stearate 18K	\	\	2	\	\
M-H/LSi	\	\	\	2	\
PTFE JTC-308	\	\	\	\	2

**Table 12 polymers-14-01795-t012:** Wear resistance of TPE prepared by various lubricating agents and masterbatches.

Sample	Taber Abrasion		Cross Scratch
	Mass Loss/mg	Appearance Rating	ΔE
TPE-no slip	0.04	4.5	0.61
TPE-HG650	0.03	6.0	0.32
TPE-SR100B	0.02	8.0	0.21
TPE-18K	0.03	6.0	0.25
TPE-(H/LSi)	0.03	8.0	0.23
TPE-PTFE	0.03	6.0	0.21

## Data Availability

The data presented in this study are available on request from the corresponding author.

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
