# Peer review of "Mechanism and Influence Factors of Abrasion Resistance of High-Flow Grade SEBS/PP Blended Thermoplastic Elastomer"

_polymers, 2022, doi:10.3390/polym14091795_

Round 1

Reviewer 1 Report

Manuscript ID

polymers-1668045

Comments on the manuscript entitled "Mechanism and Influence Factors of Abrasion Resistance of High Flow Grade SEBS/PP Blended Thermoplastic Elastomer" submitted to "Polymers".

In this study, the high flow grade SEBS/PP blended thermoplastic elastomer (TPE) suitable for injection molding skins for automobile instrument panel was prepared. The effects of SEBS’s molecular weight, styrene content in the molecule, molecular structure and lubricating system on the friction and loss properties of the material were investigated. I think this article can be accepted after some minor revisions.

  1. “Figure 3… the upper left one is the magnified view before Taber abrasion test” was miswritten. It should be “the bottom left one is the magnified view before Taber abrasion test”.
  2. The naming of the sample is confusing. For example, there are two TPE-HWMS samples. It is recommended to name the samples after the main features.
  3. The fluidity is very important for the processing of injection molding skins, and thus the corresponding data should be given.
  4. The influence of thermal stability of SEBS, entanglement between molecular chains, compatibility between SEBS and PP on the abrasion resistance of the TPEs should be concluded in the end of Section 3.
  5. The influence of lubricating system on the properties of high-flow TPE is clear, the mechanism should also be given.

Author Response

Point 1: “Figure 3… the upper left one is the magnified view before Taber abrasion test” was miswritten. It should be “the bottom left one is the magnified view before Taber abrasion test”.

Response 1: Good question! “Figure 3… the upper left one is the magnified view before Taber abrasion test” has been modified to “the bottom left one (e) is the magnified view before Taber abrasion test”.

Point 2. The naming of the sample is confusing. For example, there are two TPE-HWMS samples. It is recommended to name the samples after the main features.

Response 2: Good suggestion! The naming of the sample is modified to name the sample based on its main characteristics, such as molecular weight level, styrene content level and molecular structure. The modified samples are named TPE-Ms/Hw, TPE-Hs/Hw and TPE-Hs/Lw respectively.

Point 3: The fluidity is very important for the processing of injection molding skins, and thus the corresponding data should be given.

Response 3: Good question! Melt flow rate (MFR) was used to characterize The fluidity of materials and the relationship between MFR and fluidity of material was described in line 137-139 of the revised manuscript.

Point 4: The influence of thermal stability of SEBS, entanglement between molecular chains, compatibility between SEBS and PP on the abrasion resistance of the TPEs should be concluded in the end of Section 3.

Response 4: Good suggestion! The influence of thermal stability of SEBS, entanglement between molecular chains, compatibility between SEBS and PP on the abrasion resistance of the TPEs have been concluded and added in the end of Section 3 of revised manuscript.

Point 5: The influence of lubricating system on the properties of high-flow TPE is clear, the mechanism should also be given.

Response 5: Good suggestion! The mechanism of action of lubrication systems on the properties of high-flow TPE is as follows:By adding different lubricating agents in the formula, the friction coefficient of the material can be effectively reduced, the frictional force and the abrasion heat generation can be reduced, followed by abrasion resistance of the material can be improved significantly.

The corresponding descriptions have been added in line 333-336 of revised manuscript.

Reviewer 2 Report

In this study, the influencing factors of abrasion resistance of high flow grade SEBS/PP blended thermoplastic elastomer were studied. I think this is a valuable work, and it can be accepted after addressing following issues.

  1. The effect of molecular weight of SEBS, the content of styrene in the molecular chain, molecular structure, and the lubricating system on the friction loss resistance of the material were mainly studied. Are there any reports on the influence of these factors on the wear resistance of materials? The authors should summarize these developments in the Introduction to highlight the necessity and importance of this research.
  2. The mechanical properties of TPE, especially the tensile strength and elongation at break, should use the average value obtained after multiple measurements. The author should supplement the variance of the corresponding values. Besides, in Figures 1, 4, 7, and 11, the scale bar should be added.
  3. I think Figure 9 is the magnified photos of samples after Taber abrasion test. However, line 301~316 recommended those as discussing the result of cross-scratch test, please check the content and revise it.
  4. “environmental protection” should be changed to “environmental-friendly”. In the Line 65-66,“The elastomer material is soft, and the contact area is prone to large deformation under pressure”, “pressure” should be changed to “compression” or “compression force”.

Author Response

point 1: The effect of molecular weight of SEBS, the content of styrene in the molecular chain, molecular structure, and the lubricating system on the friction loss resistance of the material were mainly studied. Are there any reports on the influence of these factors on the wear resistance of materials? The authors should summarize these developments in the Introduction to highlight the necessity and importance of this research.

Response 1: Good suggestion! In the introduction, developments about the effects of molecular weight of SEBS, the content of styrene in the molecular chain, molecular structure, and the lubricating system on wear resistance are summarized, and the necessity and importance of this research are expounded. The corresponding descriptions have been added in the Introduction of revised manuscript (line 78-85).

Point 2: The mechanical properties of TPE, especially the tensile strength and elongation at break, should use the average value obtained after multiple measurements. The author should supplement the variance of the corresponding values. Besides, in Figures 1, 4, 7, and 11, the scale bar should be added.

Response 2: Good suggestion! standard deviation σ was used to characterized variances of tensile strength and elongation at break of 5 tested pieces (line 140-142 of the revised manuscript) and the results have been supplemented in table 2, 5, 8. Besides, in Figures 1, 4, 7, and 11, the scale bar has been added.

Point 3: I think Figure 9 is the magnified photos of samples after Taber abrasion test. However, line 301~316 recommended those as discussing the result of cross-scratch test, please check the content and revise it.“environmental protection” should be changed to “environmental-friendly”. In the Line 65-66,“The elastomer material is soft, and the contact area is prone to large deformation under pressure”, “pressure” should be changed to “compression” or “compression force”.

Response 3: Good suggestion! Figure 9 is the magnified photos of samples after Taber abrasion test, and figure 7 is the photos after cross-scratch test. Content has been revised in line 314-329 of revised manuscript. The English grammar, spelling, sentence structure, and figures have been double checked and smoothed through the whole manuscript.